

# Soil organic carbon mobility in equatorial podzols: soil column experiments

Patricia Merdy[1], Yves Lucas[1], Bruno Coulomb[2], Adolpho J. Melfi[3], Célia R. Montes[3]

[1]Université de Toulon, Aix Marseille Université, CNRS, IM2NP, 83041 Toulon CEDEX 9, France
[2]Aix Marseille Université, CNRS, LCE, Marseille, France
[3]IEE, NUPEGEL, Universidade de São Paulo, São Paulo 05508-010, Brazil

*Correspondence to*: Patricia Merdy (merdy@univ-tln.fr)

**Abstract.** Transfer of organic carbon from topsoil horizons to deeper horizons and to water table is still little documented, in particular in equatorial environments despite the high primary productivity of the evergreen forest. Due to its complexing

capacity, organic carbon also plays a key role in the transfer of metals in the soil profile and therefore in pedogenesis and for metal mobility. We were interested in equatorial podzols, which are known to play a significant role in carbon cycling. We carried out soil column experiments using soil material and percolating solution sampled in an Amazonian podzol area. The dissolved organic matter (DOM) produced in the topsoil was not able to percolate through the clayey, kaolinitic material from the deep horizons and was retained in it. When it previously percolated through the Bh material, there was production

of fulvic-like, protein-like compounds and small carboxylic acids able to percolate through the clayey material and increasing the mobility of Al, Fe and Si. Podzolic processes in the Bh can therefore produce a DOM likely to be transferred to the deep water table, playing a role in the carbon balances at the profile scale, and owing to its complexing capacity, playing a role in deep horizon pedogenesis and weathering. The order of magnitude of carbon concentration in the solution percolating in depth was around 1.5-2.5 mg L$^{-1}$.

## 1 Introduction

At the global scale, soil organic matter (SOM) constitutes the largest terrestrial reservoir of organic carbon (Hiederer and Kochy, 2012) and therefore understanding its dynamics is crucial in predicting its behavior in the context of climate and land use change. The soil carbon pool studies, initially focused on the A horizon or in 0-0.3 m depth, were subsequently extended to the upper 2 m (Batjes, 1996) or deeper (Montes et al., 2011; James et al., 2014; Pereira et al., 2016). These studies showed

that the deep soil carbon – below 0.3 m – can represent a high proportion of the total organic carbon (OC) stored in a profile (30 to 63%, Batjes, 1996). Although the interest in better quantifying the deep carbon pool and understanding its dynamics was underlined ten years ago (Rumpel and Kögel-Knabner, 2011), these points remain of interest due to a low number of existing data. In recent works, deep carbon dynamics was inferred from various types of studies, as the use of isotopic tracers such as $^{13}$C or $^{14}$C (Mathieu et al., 2015; Stahl et al., 2016; Doupoux et al., 2017; Balesdent et al., 2018; van der Voort et al.,

2019), field measurements (Lucas et al., 2012; Wan et al., 2018; Gibson et al., 2019), laboratory experiments as column



(Guo and Chorover, 2003) or respiration experiments (Fontaine et al., 2007; Lucas et al., 2020). Despite this, little is known about the fluxes and characteristics of organic matter capable of migrating in depth.

In this context, we were interested in equatorial podzols which are known to play a significant role in carbon cycling. In Amazonia, these soils store a large carbon pool, estimated around 13.6 PgC (Montes et al., 2011), and Pereira et al. (2016)

specified that they contain in average 105.9 kgC m$^{-2}$, of which 83.2 are in the deep Bh. Doupoux et al. (2017) modelled their genesis and dynamics by considering both total C fluxes and $^{14}$C fluxes. They noticed, however, that dissolved organic carbon (DOC) fluxes at depth were not enough well known to constrain the model unambiguously. The DOC fluxes exported and reaching the deep water table were generally approximated by the analysis either of groundwater taken from boreholes, of spring water baseflow at the outlet of an elementary watershed of known characteristics or by tracer-aided

modelling at the scale of a larger catchment (Birkel et al., 2020). Such data are scarce, and Table 1 summarizes those we found relating to soil systems from tropical or equatorial environments. They show that the solutions which percolate at depth in the soils of tropical or equatorial environments have significant DOC contents, varying from 0.3 to 2.3 mg L$^{-1}$. Regarding podzols, data relating to springs give information on the solutions that flows laterally in the eluvial horizons but there is very little detailed data on solutions from horizons below the Bh.


**Table 1. DOC content in deep percolating solutions.**

| Soil type | Sampling depth (cm) | DOC (mgC L$^{-1}$) | Reference |
|---|---|---|---|
| Amazonian rainforest | | | |
| Ferralsol | 450 | 1.4 ± 1.1 | McClain et al., 1997 |
| Hydromorphic ferralsol | 200 | 2.3 ± 0.9 | McClain et al., 1997 |
| Podzol | 500 | 2.3 ± 0.7 | Lucas et al., 2012 |
| Amazonian transitional rainforest | | | |
| Ferralsol | 169–965 | 1.5 ± 2.7 | Neu et al., 2016) |
| Oxisols | Springs | 0.51 ± 0.05 | Johnson et al., 2006 |
| Ultisols | Springs | 0.47 ± 0.05 | Johnson et al., 2006 |
| Oxisols | 800 | 1.0 ± 0.2 | Johnson et al., 2006 |
| Ultisols | 800 | 1.1 ± 0.5 | Johnson et al., 2006 |
| Costa-Rica rainforest | | | |
| Ultisols and inceptisols | Spring | 0.29-0.84 | Osburn et al., 2017 |
| Cameroon rainforest | | | |
| Ferralsols | Spring | 1.0 ± 0.8 | Boeglin et al., 2003 |
| Ferralsols | 800 | 1.3 ± 0.6 | Boeglin et al., 2003 |



In addition to total DOC, the characteristics of the organic species help to understand its mobility and potential degradability. Data from cambisol, acrisol or ferralsol catchments (Johnson et al., 2006; Osburn et al., 2017) showed that the

DOC percolating in depth was composed of smaller, less aromatic species than DOC from topsoil horizons. Lucas et al. (2012) obtained similar results in solutions from podzolic system that had percolated through, successively, a Bh and a 2-m thick clayey kaolinitic material.

The sampling of deep solutions, often difficult to achieve in hard-to-reach areas such as the equatorial forest, can be effectively supplemented by laboratory simulations such as column percolations, with the objective of assessing the potential

mobility of a given DOC through a soil material. This approach is part of a broader theme addressed by many authors who have been interested in interactions between natural organic matter and minerals, most often related to the mobility of contaminants, such as Hg in Amazonian podzols (Miretzky et al., 2005) . Clay minerals, through their sorption properties, promote interactions which condition the transport of DOC in soils and sediments (Jardine et al., 1989; McDowell and Wood, 1984; McCarthy et al., 1993; Kaiser and Guggenberger, 2000). The oxyhydroxides s.l. (oxides, oxyhydroxides s.s.

and hydroxides) of the soil, because of their small sizes and their surfaces with variable charge, also have strong capacities for DOC sorption. Meier et al. (1999) have shown that goethite has a higher sorption capacity than kaolinite and that, for sufficiently low DOC concentrations (<20 mgC L$^{-1}$), the adsorption of large molecules and aromatic groups is favored.

Kaiser and Zech (2000) showed that 85 to 95% of the sorption capacity of DOC by the soils they studied (alfisol and inceptisol) was due to the fine fraction (< 2 µm), and that the Fe and Al oxyhydroxides s.l. were primarily responsible for

this sorption capacity. They also showed that these minerals promoted the sorption of the more hydrophobic fraction, derived from lignin, while the clay minerals, mainly kaolinite and illite, promoted sorption of the more hydrophilic fraction. Torn et al. (1997) also showed, through the study of Hawaiian chronosequences ranging from andosols to oxisols, that the organic matter retention capacity of soil materials and the turn-over of the adsorbed SOM depended on their mineralogy. Kaolinitic clays are of special interest: they are in depth the main adsorbing mineral in high rainfall tropical and equatorial areas where

the consequences of the high primary productivity of the rainforest on the carbon cycle are already poorly understood (Grace et al., 2001).

In this framework, we carried out soil column experiments using soil material and percolating solution from an Amazonian podzol area in order to give some insight to the following questions: (1) what differences exist between the DOM that percolate in depth and the DOM produced in the topsoil, (2) does Bh impact the characteristics and fluxes of DOM

percolating in depth, (3) what are the properties of the deep percolating DOM with respect to the transport of metals involved in mineral equilibria, therefore pedogenesis?

## 2 Materials and methods

The various materials used to pack the columns were previously analyzed. The soil particle size distribution and mineralogical determinations were performed after SOM mineralization using $H_2O_2$. Particle size distribution was





determined using the Robinson pipette method. Soil organic carbon (SOC) was determined using a CHNS (FLASH 2000 Analyzer, Thermo Fisher Scientific). Kaolinite and gibbsite determination were performed by Thermogravimetry-Differential Thermal Analysis (TG-DTA) using a Shimadzu DTG-60H-Simultaneous DTA-TG. Fe-oxides were calculated as $Fe_2O_3$ after total Fe determination by ICP-AES on aqua regia digestion extracts.

The columns used were 60 cm long with an internal diameter of 3 cm. To represent the E horizon we used a quartz sand (Q)
(pure Fontainebleau sand, commercial, particle size <350 µm). To represent the other horizons, we sampled soil material in Amazonian podzol profiles: two sandy Bh material (Bh1, SOC 4.5% and Bh2, SOC 1.7%) and a kaolinitic material (K) from a horizon underlying a Bh (Table 2). All materials were passed through a 2-mm sieve and carefully packed in column to avoid large voids.

**Table 2. Characteristics of the soil material used in the columns. Weight % of bulk samples. ND: non detected.**

|       |                     |     | Fine fraction | Silt    | Fine sand  | Coarse sand |          |          | Fe     |
|-------|---------------------|-----|---------------|---------|------------|-------------|----------|----------|--------|
| Name  | Munsell color       | SOC | 0-2 µm        | 2-50 µm | 0.05-2 mm  | 0.2-2 mm    | Kaolinite | Gibbsite | oxydes |
| Bh 1  | 7.5YR2.5/1 black    | 4.5 | 2.1           | 8.8     | 60.9       | 23.7        | ND       | ND       | 0.2    |
| Bh 2  | 7.5YR3/2 dark brown | 1.7 | 1.1           | 11.2    | 54.4       | 31.7        | 1.2      | ND       | 0.1    |
| K     | 5YR8/1 white        | 1.3 | 29.3          | 67.8    | 2.9        | 0           | 91.3     | 2.2      | 0.2    |

These materials were introduced into the columns in layers of 5 cm for kaolinitic material and Bh material, and of 10 cm for sand. A 0.1 mm nylon mesh was inserted between each layer avoid mixing of the phases, at the top of the column to damp the fall of the drops, and at the base of the column to avoid suffosion of the material.

Six column experiments were conducted according to the arranged percolating device shown in Fig. 1. Two columns were packed with a Q-Bh arrangement (Q-Bh1 and Q-Bh2) to observe the DOM transfer through and from the Bh. Three columns (Q-Bh1-K(a), Q-Bh1-K(b) and Q-Bh2-K) were packed with a Q-Bh-K arrangement to observe the adsorption of the DOM issued from the Bh by the kaolinitic clay material. One column (Q-K) was packed with a Q-K arrangement to observe the direct adsorption of the DOM circulating in the E horizons by the kaolinitic clay material.

The input solution was a pH 4.1 black water taken from a spring located towards the center of a podzolic area (S 0°6'42", W 66°54'09"), corresponding to the water of the perched water-table circulating in the E horizons. It was kept around 4°C until experiment. The input solution was injected at the top of the column by a peristaltic pump to obtain a downward flow of 0.05 ml mn[-1] for 3 weeks to obtain around 1.3 L of percolate. In parallel, the peristaltic pump was used in suction mode at the output to homogenize the speed and have the same residence time in all the columns. The output solution was sampled
every 5 days, giving for each column 5 samples (fractions F1 to F5) of approximately 250 mL each. For each column, a composite sample was formed by proportional mixing of the fractions.





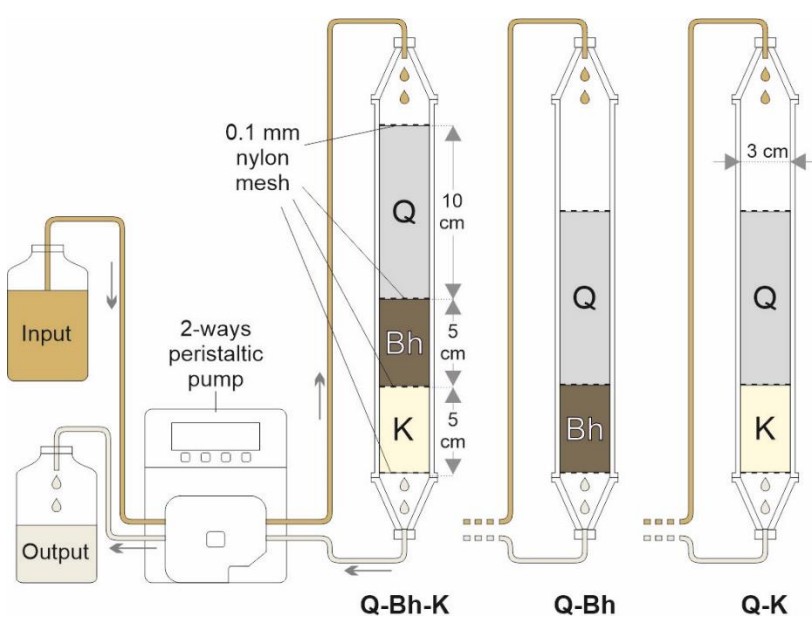

**Figure 1. Experimental setup. Q: quartz sand; Bh: sandy Bh material; K: clayey kaolinitic material.**

The input solution as well as the percolates were filtered on a 0.45 µm cellulose nitrate filter previously washed with milliQ water and on which blanks made it possible to check the absence of contamination regarding the elements analyzed. DOC, Fe and Al concentrations were determined for each fraction; Si and small organic acids (SOA) (formic, oxalic, malic ...) were determined on composite samples.

    Input solution and percolates were analyzed according to the following techniques. The DOC was determined by TOC-
meter (TOC-V, SHIMADZU) coupled to an ASI-V automatic sampler. The dissolved organic matter (DOM) was characterized by 3D fluorescence (Excitation Emission Matrix Fluorimetry, EEMF) (HITACHI F-4500 spectrometer), this method allowing a rapid characterization of fluorophores associated with humic matter and proteins (Chen et al., 2003; Nebbioso and Piccolo, 2013). Major anions and cations were determined by ion chromatography (Dionex DX 120), using 9 mmol $L^{-1}$ NaHCO$_3$ for cation elution and 10 mmol $L^{-1}$ methane sulfonic acid for anion elution. Si, Al, Fe were quantified by
ICP-AES. SOA (formic, oxalic, malic ...) were determined and quantified by high performance ion chromatography (Dionex ICS-3000) coupled to a mass spectrometer (MSQ Plus, Thermo Scientific) driven by Chromeleon® (6.80 version) and equipped with an AG11-HC guard column (Dionex), an IonPac AS11-HC column (4x250 mm, Dionex) and using a 25µL loop injection valve. Analysis were performed in a gradient mode (from 1 to 5 mM NaOH in helium sparged demineralized water in 40 min.) at 30 °C, with a flow rate set at 0.8 mL.min-1. To improve the signal-to-noise ratio of the measurement, an
external flow electrochemical suppressor system (ACRS 500 4 mm) was added to the analytic system.



# 3 Results and discussion

## 3.1 Transfers of carbon and major elements

The pH of the percolating solution was 4.0, it was not modified by passing through the columns. For each column, the variations in DOC concentration between fractions throughout the experiment remained much lower than the differences
between columns and did not exhibit clear trends (Fig. 2); the same evolution was observed for Al and Fe concentrations. There were therefore no significant changes within a given column experiment in the behavior of the columns which allows us to discuss the results using the composite sample compositions given in Table 3.

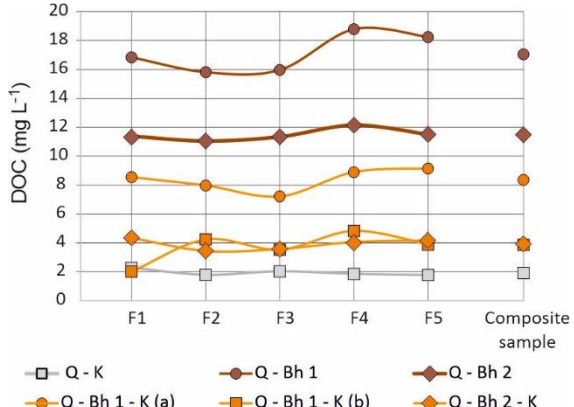

**Figure 2. DOC in the five fractions (F1 to F5) of the column percolate and in the composite sample. Analytical errors are smaller than the point marks.**

We can consider that the output of the Q-Bh type columns corresponded to the input into the kaolinitic material of the Q-Bh-K type columns, as sketched in Fig. 3. If there was a relatively large variation between columns of the same type, the
differences between columns remained consistent of different types remained consistent.





**Table 3. C, Al, Fe and Si transferred in dissolved phase in the column experiments. Output values correspond to the composite samples. Analytical percent error: C, 1.4; Al, 6.0; Fe, 4.8; Si, 1.9. na: not analyzed.**

|  | Q-K | Q-Bh1 | Q-Bh2 | Q-Bh1-K (a) | Q-Bh1-K (b) | Q-Bh2-K |
|---|---|---|---|---|---|---|
| DOC concentration in input (mg L$^{-1}$) | 31.4 | 31.4 | 31.4 | 31.4 | 31.4 | 31.4 |
| DOC concentration in output (mg L$^{-1}$) | 1.9 | 17.1 | 11.5 | 8.4 | 3.9 | 3.9 |
| Al concentration in input (µg L$^{-1}$) | 22 | 22 | 22 | 22 | 22 | 22 |
| Al concentration in output (µg L$^{-1}$) | 17 | 658 | 490 | 286 | 91 | na |
| Fe concentration in input (µg L$^{-1}$) | 61 | 61 | 61 | 61 | 61 | 61 |
| Fe concentration in output (µg L$^{-1}$) | 6 | 153 | 32 | 95 | 62 | 3 |
| Si concentration in input (µg L$^{-1}$) | 287 | 287 | 287 | 287 | 287 | 287 |
| Si concentration in output (µg L$^{-1}$) | 585 | 583 | 393 | 891 | 953 | 1182 |


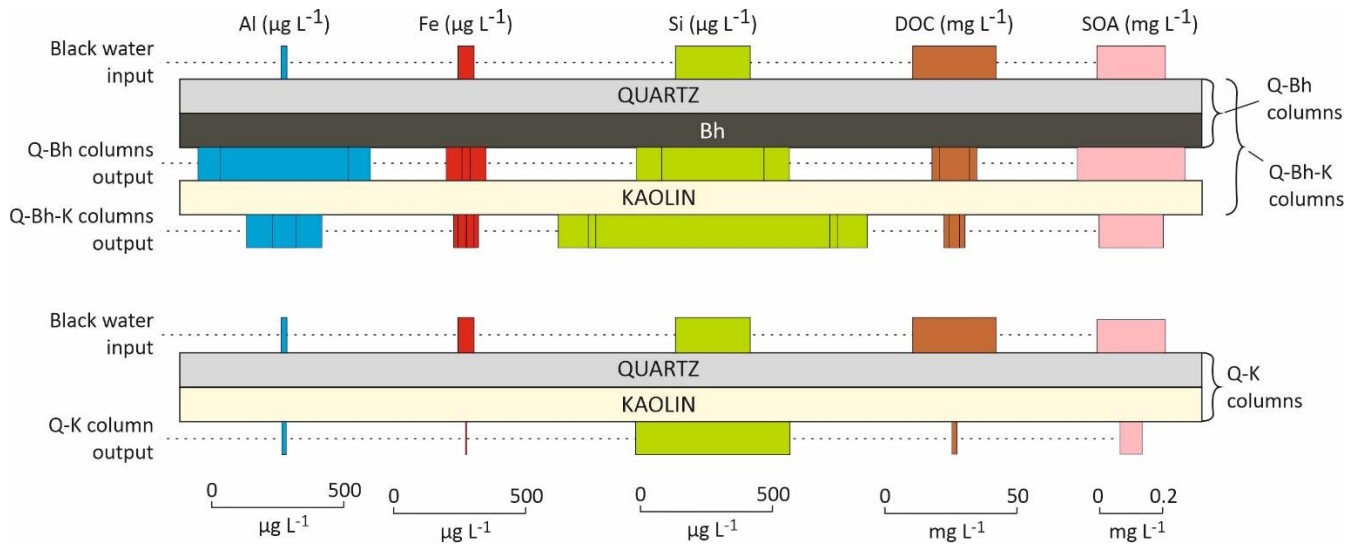

**Figure 3. Representation of the concentrations measured during the column experiments. The width of the coloured blocks represents the concentration of the solute. SOA: small organic acids.**

Regarding DOC, the lowest column outlet concentration (1.9 mg L$^{-1}$) was observed for the Q-K column (Table 3), where 94% of the carbon introduced at the top of the column was retained; consistently, the upper part of the kaolinitic layer acquired a light brown color. The highest DOC outlet concentrations (17.1 and 11.5 mg L$^{-1}$) were observed for the two Q-Bh type columns, which was expected but where, however, 46 and 63% of the carbon introduced were retained. The DOC concentrations at the outlet of the three Q-Bh-K type columns were intermediate (8.4; 3.9 and 3.9 mg L$^{-1}$), corresponding to

the retention of 81, 88 and 89% of the introduced carbon, respectively. In summary, the Bh retained a part of the introduced





DOC and the kaolinitic material retained most of the DOC of the solution which percolated through. The presence of a Bh, however, increased the proportion of carbon which passed through the kaolinitic material. The nature of the DOM released by the Bhs was therefore probably different from that of the DOM of the input, i.e. of the black water from the perched water table; this was likely due to a specific microbial activity within the Bh in the columns.

Regarding Si, the column outlet concentrations were always higher (393 to 1182 µg L$^{-1}$) than the inlet concentration (287 µg L$^{-1}$): all columns released Si. Percolation by the input solution through the Bh only (Q-Bh), or the kaolinitic material only (Q-K), released Si in similar proportions (outlet concentrations from 393 to 585 µL$^{-1}$); the greatest releases (891, 953 and 1182 µg L$^{-1}$) were observed after successive percolation through Bh and kaolinitic material in the Q-Bh-K type columns.

        Aluminum exhibited a very different behavior from that of Si. Percolation through the Bh (Q-Bh) by the input solution
(Al concentration 22 µg L$^{-1}$) resulted in high release of Al (490 and 658 µg L$^{-1}$), when percolation through the kaolinitic material only (Q-K) resulted in a partial retention of Al (Al concentration at outlet 17 µg L$^{-1}$). As for DOC, however, the previous percolation through the Bh (Q-Bh-K columns) increased the proportion of Al which passed through the kaolinitic material. The discrepancy between Si and Al behavior shows that these elements were not controlled by a congruent dissolution of kaolinite. Iron concentration pattern was quite similar to Al, but with lower concentrations at column outlets.

The behavior of Fe and Al could be explained by the release either of OM-complexes (organo-metallic complexes) (Lucas, 2001; Patel-Sorrentino et al., 2006), or mineral colloids (kaolinite, gibbsite, goethite) (Cheng and Saiers, 2015) during percolation through the Bh, these complexes or colloids being subsequently partially retained during percolation through the kaolinitic material. In the Bh output, Si and Al were released in a stoichiometry equivalent to that of kaolinite, which would be compatible with a release of colloidal kaolinite. Recent studies, however, suggested that Si and Al can be
transferred as ternary OM-Al-Si complexes (Merdy et al., 2020). Anyway, the behavior of these two elements during the subsequent percolation through the kaolinitic material diverged completely: the kaolinitic material retained Al while it released Si, suggesting that Al was released by the Bh as OM-complexes. This hypothesis is strongly supported by the high correlation between DOC and Al (Fig. 4). Fe behaved similarly, but with a weaker correlation with DOC, which suggests that Al was only transferred as DOM-Al complexes when Fe could also be transferred as mineral colloids.






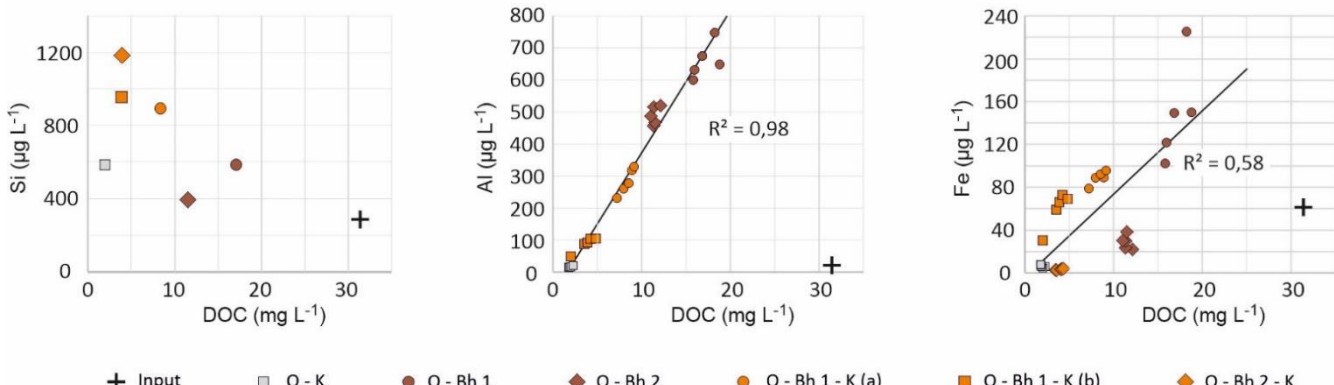

**Figure 4. Relationships between DOC and dissolved Al, Fe and Si in column percolates. The cross represents the concentration in the input solution. Values are from composite samples for Si and from individual fractions for Al and Fe.**

To assess the consistency of these interpretations, we built a Si-Al diagram considering the data obtained from the studied solutions, taking into account metal complexation by the DOM in one hand, or not taking it into account in the other hand, (Fig. 5). The parameters of the dissolved organic matter necessary for the quantification of the complexation were a site density equal to 27 µmol mg$^{-1}$ (Lucas et al., 2012) and a DOM-Al conditional stability constant equal to $10^5$ (Lee, 1985; Hagvall et al., 2015). The "Kaolinite 1" line corresponds to the stability of kaolinite calculated with the WATEQ4F database,

which uses a solubility product (Ksp = $10^{3.705}$) identical to that proposed by Tardy and Nahon, (1985) after a critical analysis of the literature. The "Kaolinite 2" line uses the solubility product (Log (Ksp) = $10^{2.853}$) proposed by Grimaldi et al. (2004) to report on equatorial supergene kaolinites substituted in iron and of variable crystallinity.

When the DOM-Al complexes are not considered, Q-Bh1, Q-Bh2 and Q-Bh1-K(a) output solutions seems near equilibrium with kaolinite. However, when DOM-Al complexes are considered, all output solutions are far undersaturated

with kaolinite and gibbsite. These solutions, which remain aggressive towards the kaolinite, are therefore able to dissolve the upper part of the horizons located under the Bh, promoting the thickening of the profile.





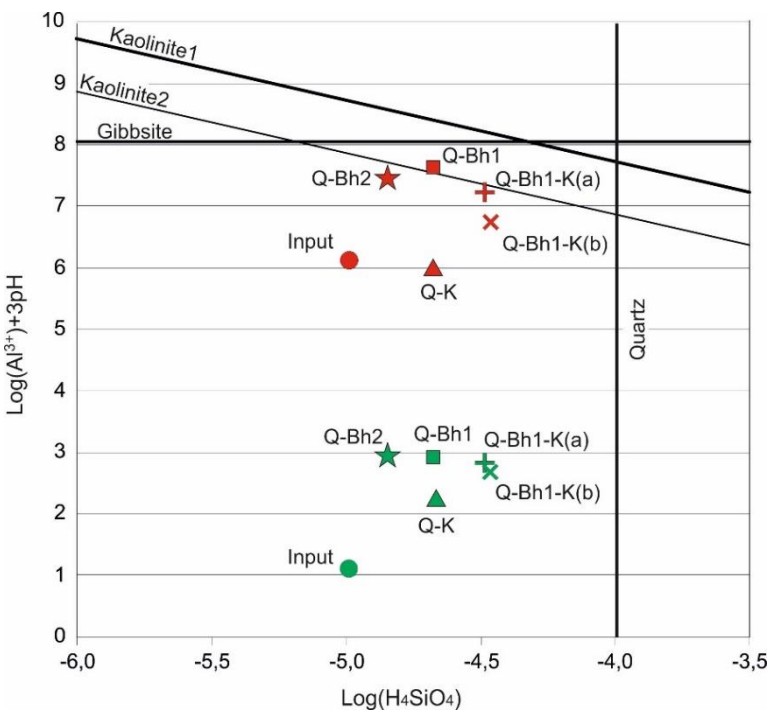

**Figure 5. Situation of the studied solutions in the Si-Al system. Red symbols: without considering DOM-Al complexes; green symbols: considering DOM-Al complexes.**

### 3.2 Fluorescence properties of percolation solutions

Fluorescence spectroscopy is an appropriate tool to characterize natural organic matter whose fluorophores give specific signals. Excitation-emission fluorescence matrix (EEFM) of input and output solutions are given in Fig. 6. The peak A corresponds to fulvic-like humic compounds, the peak C to humic-like humic compounds, the peak P to protein-like compounds that indicate an active bacterial activity (Coble, 1996), these peaks being characteristics of natural terrestrial DOM. The peaks S1 and S2 have been related to non-humic like, labile matter related to microbial activity (Singh et al., 2010) or to fulvic-like compounds (Stedmon and Markager, 2005).





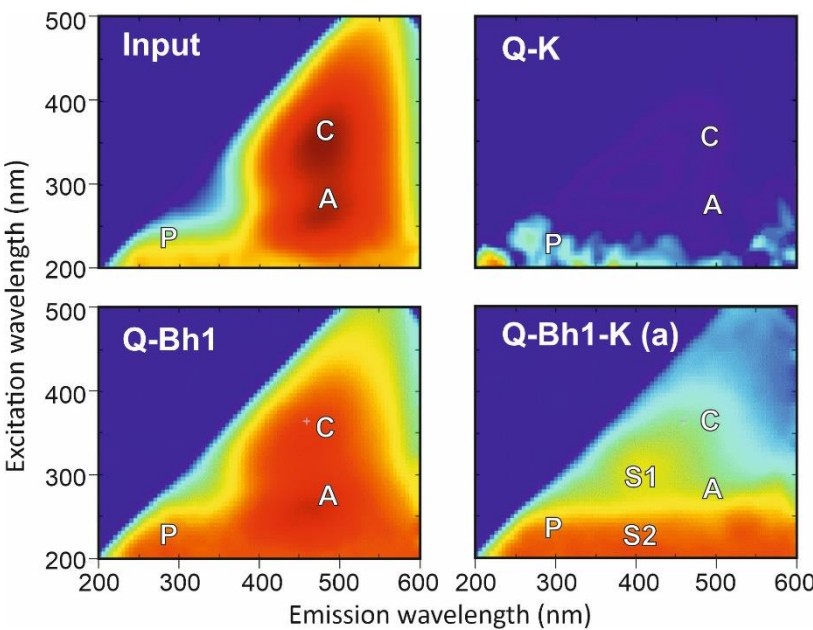


**Figure 6. Excitation-emission fluorescence matrix of the solutions, arbitrary units. The letters identify the usual position of the peaks P (protein-like), C (humic-like), A (fulvic-like), S1 and S2 (non-humic or fulvic like).**

The EEFM of the input solution was typical of humified DOM with a dominant C peak, a marked A peak and a small P

peak. After the input solution has passed through kaolinitic material (Q-K output), there is hardly any humified DOM and a very reduced signal of the protein-like DOM: nearly all fluorescent DOM was retained in the kaolinitic material. After the input solution has passed through the Bh (Q-Bh1 output), there was a reversal of the C/A intensity ratio, which indicates a partial retention of the most condensed DOM in the Bh, an a higher P peak, which indicates a bacterial activity within the Bh. After the input solution has passed through both Bh and kaolinitic material (Q-Bh1-K(a) output), the DOM exhibited

some humified character, more fulvic than humic, and protein-like feature.

These observations confirmed that the DOM released by the Bh was different from the DOM of the input solution. The Bh retained the most humified DOM compounds and released compounds capable of being transferred through clay materials, more fulvic-like or protein-like as issued from active bacterial activity.

**3.3 Transfer of small organic acids**

Chromatography was used to identify the composition of the DOM present in the column experiments. Lactic and malic acids were the only small organic acids found in the input solution at detectable concentrations (Table 4). After the input solution has passed directly through kaolinitic material (Q-K output), 68% of the carbon from the measured SOA was retained in the kaolinitic material and it only remained a low concentration of lactic acid. Percolation of the input solution through the Bh (Q-Bh output) resulted in an increase in the quantities and variety of SOA, indicating microbial activity





during the experiment, which is consistent with fluorescence observations. Comparing Q-K and Q-Bh-K outputs shows that a previous percolation through the Bh increased, as for DOC, the proportion of SOA carbon which passed through the kaolinitic material (Fig. 3). This observation is consistent with fluorescence data and strengthened the hypothesis that the DOM released by the Bhs was different from that of the input solution.

**Table 4. Concentrations ± standard deviation of small carboxylic acids in the studied solutions and from literature. Values in mgC L-1, <dl: lower than detection limit.**

|  | Input solution | Q-Bh-K output (8 samples) | Q-Bh output (3 samples) | Q-K output (2 samples) | Water table beneath a kaolinitic horizon[a] |
|---|---|---|---|---|---|
| Formic acid | <dl | <dl | <dl | <dl | 0.44 |
| Acetic acid | <dl | <dl | <dl | <dl | 0.10 |
| Propionic acid | <dl | <dl | <dl | <dl |  |
| Lactic acid | 0.196 | 0.065 ± 0.020 | 0.164 ± 0.028 | 0.070 ± 0.042 | 0.02 |
| Oxalic acid | <dl | 0.029 ± 0.025 | 0.110 ± 0.065 | <dl | 0.09 |
| Valeric acid | <dl | <dl | <dl | <dl |  |
| Malic acid | 0.021 | 0.007 ± 0.005 | 0.030 ± 0.004 | <dl |  |
| Citric acid | <dl | 0.075± 0.082 | <dl | <dl | 0.07 |
| Succinic acid | <dl | 0.028 ± 0.024 | 0.038 ± 0.001 | <dl |  |
| Total SOA C | 0.217 | 0.204 | 0.341 | 0.07 |  |

[a]Lucas et al. (2012)

### 3.4 From experiment to field

To what extent can the conclusions of the experiment be extrapolated to field conditions? The column experiments exhibited
differences with field usual conditions.

lasted only 3 weeks, when at field under usual conditions a quasi-permanent water table is perched over the Bh which has a low hydraulic conductivity. The solution circulating in the E horizon likely percolates very slowly in the Bh throughout the year (Ishida et al., 2014). In the columns the Bh was reworked, which ensured a higher hydraulic conductivity and most likely different soil-solution contact conditions from that at field, and was previously dried out, which may result in a change
in microbial activity (Denef et al., 2001). Microbial activity is also sensitive to redox conditions. Our column experiments were conducted without control of the redox potential, when deep *in situ* Bh can be submitted to reducing conditions (Lucas et al., 2012).

Nevertheless, the column experiment showed negligible variations with time of the percolate characteristics, which suggests a steady state. Results are also consistent with the scarce field data available:



• The ratio of input/output average DOC concentration for the Q-Bh-K columns ranged from 0.12 to 0.27, which is in the range of those observed between DOC concentration in E horizons and deep water table (0.13, Lucas et al., 2012), or predicted by podzol genesis modelling (0.14 to 0.35, Doupoux et al., 2017).

• In the column experiments, the DOM that percolates through the kaolinitic material had a higher content of small carboxylic acids and of fulvic-like compounds, i.e. less aromatic than DOM of the input solution. This is consistent with
the observations of Lucas et al. (2012), who have found that the DOC of the water table situated under a kaolinitic horizon in a podzolic area had a high proportion of small organic compounds with high complexing capacity.

It is therefore possible to conclude that percolation through the Bh plays a key role in the geochemistry of the system, by producing compounds able to transfer both DOC and metals through kaolinitic materials.

## 4 Conclusion

The column experiments led to conclusions shown schematically in Fig. 7. The DOM produced in the acidic upper horizons and circulating in the E horizons would be highly adsorbed, with a complete retention of the humified compounds, if directly percolating through a clayey, kaolinitic material. If this DOM percolates previously through a Bh, it is subjected to transformation in this horizon. The humic-like compounds are retained, and a more fulvic-like, proteinaceous DOM containing small organic acids, which is more likely to percolate through a kaolinitic material, is released.

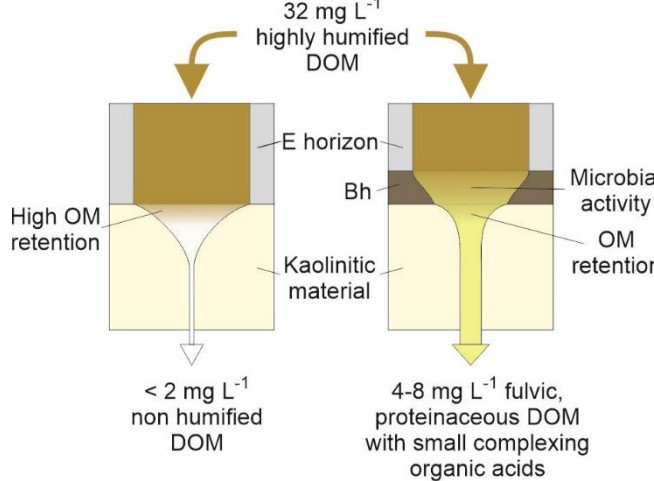


**Figure 7. Main conclusions diagram.**

The DOM that percolates in deep horizons is therefore different that the highly humified DOM that circulates in the E horizon. Microbial activity in an *in situ* Bh may be different from that observed in the columns, but the input/output C ratio
of our experiments was in the range of what has been observed in field or predicted by modelling. A DOC concentration around 1.5-2.5 mg L$^{-1}$ for solutions percolating through deep kaolinitic horizons appeared therefore as a good order of



magnitude. The higher proportion of small organic acids in the solution able to percolate through deep kaolinitic horizons confirmed its ability to transfer metals such as Al or Fe as organo-metallic complexes, increasing therefore the leaching in depth of these elements.

These conclusions strength the hypothesis given in Ishida et al. (2014) related to the genesis of tropical podzols. The solution that percolates through the Bh is able to transfer metals through a kaolinitic material, therefore to promote the downward progression of the E/Bh horizons by weathering the upper part of the kaolinitic deep horizons.

**Data availability**. The data used in this study are available from the corresponding author.


**Author contributions.** PM designed the experiments that were carried out by PM and YL. YL, CRM and AJM did the field sampling. BC realized the SOA analysis. PM and YL prepared the manuscript with contributions from all the co-authors. YL and CRM were project leaders in France and Brazil, respectively.

**Competing interests**. The authors declare that they have no conflict of interest.

**Financial support.** This work was supported by the FAPESP (São Paulo Research Foundation) [grant numbers: 2011/03250-2, 2012/51469-6]; the CNPq (Brazilian National Council for Scientific and Technological Development) [grant numbers: 303478/2011-0, 306674/2014-9]; the French ANR (French National Research Agency) [grant number ANR-12-
IS06-0002 "C-PROFOR"].

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
