# Peer review of "Soil organic carbon mobility in equatorial podzols: soil column experiments"

_SOIL, 2021_

## Referee Comment (RC2)

[referee-annotated manuscript omitted]

---

## Author Response (AR1)

**Soil-2021-3 manuscript**

**Response to the Referee 1 comments (hereafter in italics)**

*Comment: Could you add the work's reference where those Bh horizons were sampled, we don't need the coordinates, by the way it's important to express the extension of tropical podzols. Do we have to understand that the soils materials correspond to the Lucas et al 2012 reference?*

Response: Soil materials as well as percolating water come indeed from the site studied in Lucas et al., 2012. We will add this clarification by substituting the first sentence of §2 by the following: " The soil materials used to pack the columns and the percolating solution were sampled in the area described in Lucas et al. (2012) and Ishida et al. (2014), situated near the São Gabriel da Cachoeira city, Amazonia State, Brazil."

*Comment: The input water came from a field sampling, how OM and Fe were maintained in solution, was the natural pH 4.1 and the cool temperature sufficient?*

Response: From the beginning of the 2000s, we evaluated the conservation of solutions from soil and river from podzolic areas, between the time of sampling and the laboratory. We have found that storing solutions as described in the article does not alter the 3D spectra of organic matter, whereas freezing is not suitable. These solutions are very poor in nutrients and we have never observed the development of microorganisms during storage. For better clarity, we will add the following sentence: "It was kept in gas-tight plastic bottles, protected from light and around 4°C until experiment."

*Comment: We know that the Bh horizon porosity is quite limited; in general the granular porosity of the E horizon was filled up by the OM deposited by successive coating on sand grains, finally in an "old Bh", the only porosity that remains is like fissury structure revealed by micromorphology, and expressed during the possible dry period, when the water table remains only in the kaolinitic mantle, and the uppers horizons can ± dry. So please: comment in your point of view the choice you made for the sieved materials (here you enhance a macro-granular porosity) and also the value of the water flow in the columns.*

Response: We do not know the value of the water flow through the Bh at field. We failed to collect sufficiently large and undisturbed Bh samples. As the waterlogged sandy E horizons have poor cohesion, it is impossible without large means (for example installing blade planks then pumping water) to sample in a trench. Sampling had to be done in a boring hole by hammering a sampling cylinder in a very hard Bh. Doing this results in the formation of cracks in the sample. Rather than risking percolation of water in anthropogenic cracks, we preferred sieving and packing to ensure a good contact between the column material and the percolating solution.

*Comment: Did you air-dry the soil samples before sieving?*

Response: No, because we know that drying the samples may induce a change in respiration that indicates a change in microbial populations (Lucas et al., 2020). For better clarity, we will add the following sentence: "Between the sampling time and the experiment, the Bh and the kaolinitic materials were kept at field moisture and around 4°C (Lucas et al., 2020)."

*Comment: Additionally, you had a rapid comment on the possible microbial activity; can you describe the temperature conditions of the experience?*

Response: At the end of the "Materials and methods" section, we will add the following sentence: "The experience was conducted at room temperature (24°C), close to the average temperature in the forest which is around 25°C (Salati and Marques, 1984)."

*Comment: I understand that we consider that the Q-K column is a reference; I would have made a column only with the Q to express a control, and in particular to have a control with the Si, but also for the DOC. I think that you made here also a choice?*

Response: Yes, we assumed that the exchange surfaces and the dissolution kinetics of quartz sand were negligible at the time scale of the experiment. Remember also that the percolating solution was taken from the E sandy quartz material.

*Comment: The results. They are very interesting. In my first reflexion, I was expecting that the procedure used for the samples promote a "liberation" of OM from the Bh, and an elevated values of DOC.*

*Comment: Please, in the figure 4 indicate the significance of the DOC relation with Al.*

Response: We will do that, but is this really necessary? P-value are $2 \cdot 10^{-20}$ and $1 \cdot 10^{-6}$ for DOC-Al and DOC-Fe, respectively.

*Comment: I'm not sure that you have to stress a global linear relation Fe=f(DOC), your figure showed three facts: there is a positive relation in the columns Q-Bh1-K, the Q-Bh2 had another comportment, and the QzBh1 and Bh2 another one, so it seems that the behaviour of iron is complicated and regulated by the fine nature of the two Bh samples? If the iron migrated as mineral colloids, I'm not sure that it is explicated here (those colloids been stopped in the 0.45 μm filters).*

Response: Colloids can be smaller than 0.45 μm. Patel-Sorrentino et al. (2007) have shown using TFF that about 50% of the iron in a spring blackwater issued from Amazonian podzols was contained in the fraction comprised between 5 kD and 0.2 μm.

*Comment: Do you have some explanation for the differential comportments of the Bh1 and Bh2?*

Response: We propose to comment the differences between Bh1and Bh2 with regard to Fe release by adding the following sentence: " Iron and Al release was different between Bh1 and Bh2, which may be due to different crystallochemistry of Al- and Fe-bearing minerals."

*Comment: I understand the interest for the fig 5. Do you mean, sentence lines 195-196, that it is the "promotion" of the podzol volume?*

Response: Yes

*Comment: So, where was going the Si? In all columns the concentrations showed a Si-"exportation". This fact needs a return of discussion in the section 3.4 /*

Response: Results from Ishida et al. (2014) showed that formation of the kaolinitic horizons beneath the Bh is characterized by a net loss of Si, and that the podzolic processes can enhance kaolin genesis and bleaching by supplying Al and encouraging Fe leaching.

Response: We will address this question in section 3.4 by adding a bullet point: "Field data suggest that the lower boundary of the kaolinitic horizons beneath the Bh moves downward with time (Ishida et al., 2014). This is consistent with the transfer of DOM-Al complexes which can release Al in depth, promoting kaolinite precipitation. Iron can be transferred in the same way, but does not accumulate due to more reducing conditions in depth."

*Comment: Concerning the carbon-water concentration and the kaolinitic mantle, are those concentrations compatibles with specific surface of the mineral phase (retention with no-Bh columns)? we understand that a considerable part was retained.*

Response: An accurate response to this question would need complementary studies. It is, however, possible to give an order of magnitude. The weight of the K material in a column was about 80 g. Adsorption of natural organic matter on kaolinite can be evaluated from experimental studies. For example Chen et al. (2017, http://dx.doi.org/10.1016/j.jcis.2017.05.078) found 342 mg $g^{-1}$ for humic acid (HA) adsorption capacity on kaolinite at pH 4. With such a value, 80 g of kaolinitic material at 90% of kaolinite would be able to retain 24 g of HA, i.e. 300 mg $g^{-1}$. Here the DOC retained during the experiment was around 0.45 mg $g^{-1}$, a negligible value compared to the retention capacity.

*Comment: Can we suppose that the upper part of the kaolinitic mantle have to be saturated by the small molecules of OM (low molecular mass) to after observe the process of weathering of the kaolinite? Do you can link better this part of the discussion in the 3.4 section with others results. Papers on experimental pedology are scarce, and it is interesting to enhance the results.*

Response: No, to the contrary we think that SOA are poorly retained by the kaolinitic material. We will add this observation in the 3.4 section by adding the sentence " Such compounds are therefore poorly retained by the kaolinitic material."

*Comment: With regard to the carbon concentrations in water, [DOC], can you extend the discussion to the basin, or perhaps a part off (in the introduction or in the discussion), some papers where achieved on the water-carbon in the Amazonian basin.*

Response: In section 3.4 we already stated that "The ratio of input/output average DOC concentration for the Q-Bh-K columns ranged from 0.12 to 0.27, which is in the range of those observed between DOC concentration in E horizons and deep water table (0.13, Lucas et al., 2012), or predicted by podzol genesis modelling (0.14 to 0.35, Doupoux et al., 2017)." We think that extrapolation to the basin scale would need a heavy review that will be the subject of a future synthesis on equatorial podzols.

*Comment: About the last sentence in the conclusion, others works in others part of the Amazonian shields showed the development of podzols on kaolinitic soil as a result of modification of vertical drainage and then clay weathering.*

Response: We agree. As already stated in various publications from some of the authors, a set of processes drive the genesis of tropical podzols, as hyperacidity due to a negative alkalinity of the

soil solutions, the onset of a contrast in hydraulic conductivity in depth, etc. To clarify, we propose to change the first sentence of the last § as follows: "These conclusions strength the hypothesis given in Ishida et al. (2014) related to **one of the processes that drive** the genesis of tropical podzols."

*Comment: - Line 79, its not mineralization, but with H2O2 a total OM oxidation.*

Response: OK, we will substitute "mineralization" by "oxidation"

*Comment: - Please review the sentence lines 139-140, which remains unclear.*

Response: We will rewrite as follows: " In Fig. 3, we considered that the output of the Q-Bh type columns corresponded to the input into the kaolinitic material of the Q-Bh-K type columns. There was a relatively large variation between columns of the same type; the differences between columns of different types, however, remained consistent."

*Comment: - There is a problem in the edition of the sentences from line 241 to the Ishida ref.*

Response: The second sentence will be corrected as follows: "It lasted only 3 weeks, when at field …"

**Response to the Referee 2 comments (hereafter in italics)**

*General comments*
*Abstract: please state the overall aim/objective of the study and perhaps a short sentence on the overall implications of the findings.*
Response: We reworked the introduction as follows: "Transfer of organic carbon from topsoil horizons to deeper horizons and to the water table is still little documented, in particular in equatorial environments despite the high primary productivity of the evergreen forest. Due to its complexing capacity, organic carbon also plays a key role in the transfer of metals in the soil profile and therefore in pedogenesis and for metal mobility. We were interested inAs equatorial podzols, which are known to play an significant important role in carbon cycling, . Wwe carried out soil column experiments using soil material and percolating solution sampled in an Amazonian podzol area in order to better constrain the conditions of transfer of organic carbon at depth. The dissolved organic matter (DOM) produced in the topsoil was not able to percolate through the clayey, kaolinitic material from the deep horizons and was retained in it. When it previously percolated through the Bh material, there was production of fulvic-like, protein-like compounds and small carboxylic acids able to percolate through the clayey material and increasing the mobility of Al, Fe and Si. Podzolic processes in the Bh can therefore produce a DOM likely to be transferred to the deep water table, playing a role in the carbon balances at the profile scale, and owing to its complexing capacity, playing a role in deep horizon pedogenesis and weathering. The order of magnitude of carbon concentration in the solution percolating at depth was around 1.5-2.5 mg L-1. Our findings reveal a fundamental mechanism that favors the formation of very thick kaolinitic saprolites."

*Throughout:*

- *you refer to a number of hypotheses, but these are neither explicitly laid out nor rigorously tested. It would be better to replace the term hypothesis with the 'notion' or 'suggestion' or 'interpretation'*

Response: We agree with the referee and we replaced "hypothesis" by "interpretation"

- *for clarity I would refrain from using 'significant' unless based on statistical testing*

Response: We agree and we replaced significant by "important", "high" or "substantial"

*Conclusion: would you be able to add a sentence or two on the wider implications of the findings?*

Response: We added the following: "More widely, our findings reveals a fundamental mechanism that favors the formation of very thick kaolinitic saprolite where pedogenesis could act for sufficient time."

**Detailed comments**

Response: We have taken into account all the small detail corrections pointed out by the referee. Hereafter are our corrections to the comments which required a significant modification of the text.

*Line 39: meaning? please rephrase or explain*

Response: We rephrased as follows: " The DOC fluxes exported and reaching the deep water table were generally approximated (1) by the analysis of groundwater taken from boreholes or springs situated at the outlet of an elementary watershed of known characteristics, or (2) by tracer-aided modelling at the scale of a larger catchment (Birkel et al., 2020). "

*Line 175: these two sentences do not link well. please edit.*

Response: We substituted the phrase beginning with "Anyway…" by the following: "During the subsequent percolation through the kaolinitic material, the behavior of these two elements diverged completely: the kaolinitic material retained Al while it released Si, suggesting that Al was released by the Bh as OM-complexes.

*Line 242: Incomplete sentence and two issues mixed up.*

Response: We rephrased as follows: "To what extent can the above findings be extrapolated to field conditions? The column experiments exhibited differences with typical field conditions. It lasted only 3 weeks, when at field under typical conditions a quasi-permanent water table is perched over the Bh which has a low hydraulic conductivity. …"

---

## Author Response (AR2)

**Soil-2021-3 manuscript**

**Autor response to editor**

Dear editor,

We agree with your comments and we corrected the manuscript as follows, your comments are in italics:

*Line 292: replace 'strength' with 'strengthen'*
This was done

*294: replace 'to promote' with 'promoting'*
This was done

*295: replace 'reveals' with 'reveal'*
This was done

*296: last words unclear. Do you mean 'were pedogenesis acts for a sufficiently long time'?*
You are right, we corrected the sentence as suggested.